# POLICY GRADIENTS INCORPORATING THE FUTURE

**David Venuto[1,2], Elaine Lau[2], Doina Precup[1,2,3], Ofir Nachum[4]**
[1]Mila, [2]McGill University, [3]DeepMind, [4]Google Brain
`david.venuto@mail.mcgill.ca`

## ABSTRACT

Reasoning about the future – understanding how decisions in the present time affect outcomes in the future – is one of the central challenges for reinforcement learning (RL), especially in highly-stochastic or partially observable environments. While predicting the future directly is hard, in this work we introduce a method that allows an agent to "look into the future" without explicitly predicting it. Namely, we propose to allow an agent, during its training on past experience, to observe what *actually* happened in the future at that time, while enforcing an information bottleneck to avoid the agent overly relying on this privileged information. Coupled with recent advances in variational inference and a latent-variable autoregressive model, this gives our agent the ability to utilize rich and *useful* information about the future trajectory dynamics in addition to the present. Our method, Policy Gradients Incorporating the Future (PGIF), is easy to implement and versatile, being applicable to virtually any policy gradient algorithm. We apply our proposed method to a number of off-the-shelf RL algorithms and show that PGIF is able to achieve higher reward faster in a variety of online and offline RL domains, as well as sparse-reward and partially observable environments.

## 1 INTRODUCTION

Fundamentally, reinforcement learning (RL) is composed of gathering useful information (*exploration*) and assigning credit to that information (*credit assignment*). Both of these problems present their own unique learning challenges. In this work, we focus on credit assignment, which refers to the challenge of matching observed outcomes in the future to decisions made in the past. Humans appear to do this in a sample efficient manner (Johnson-Laird, 2010), and so it is natural to expect our own RL agents to do so as well.

One of the most popular approaches to credit assignment, known as *model-free* RL, is to learn a value function to approximate the future return given a starting state and action. The value function is learned using experience of the agent acting in the environment via temporal difference (TD) methods (Sutton, 1988), which regress the value function to a target based on a combination of groundtruth returns achieved in the environment and the approximate value function itself. The need to *bootstrap* learning of the value function on its own estimates is known to lead to difficulties in practice, where one must achieve a careful balance between bias and variance (Harutyunyan et al., 2019; Weaver & Tao, 2001; Schulman et al., 2016; Mnih et al.). If a slight imbalance arises, the consequences can be disastrous for learning (Tsitsiklis & Van Roy, 1996; van Hasselt et al., 2018; Sutton & Barto, 2018). For example, in offline RL this issue is so pronounced that algorithms must apply strong regularizations on both learned policy and value function to achieve stable performance (Wu et al., 2020; Kumar et al., 2019; Zhang et al., 2021; Nachum et al., 2019).

The model-free approach plays dual to the *model-based* approach, where an agent learns a dynamics and reward model of the environment, and then learns an agent to optimize behavior in this model. Thus, credit assignment boils down to utilizing an appropriate planning algorithm that can perform multiple rollouts in the model, effectively allowing the agent to "look into the future" (Racanière et al., 2017; Pascanu et al., 2017) to determine cause-and-effect (Sutton, 1991; Peng et al., 2018; Abbas et al.). While model-based RL may appear more straightforward, learning an accurate model is a challenge in practice, presenting its own sample-efficiency problems (Wang et al., 2020) as well as memory and computational issues (Łukasz Kaiser et al., 2020). Model-based approaches are thus most beneficial when the environment exhibits some level of regularity (Fra, 2019).

Beyond these issues, credit assignment in both model-free and model-based RL is further exacerbated by *partially observable* environments, in which the full environment state is not known to the learning agent. Thus it is infeasible to predict future events accurately. When applied to such non-Markovian domains, model-free algorithms relying on bootstrapping and value function approximation tend to be biased (Singh, 1994). On the other hand for model-based approaches, learning an accurate dynamics model in such domains is a difficult, potentially ill-defined problem (Suematsu & Hayashi, 1999; Bush & Pineau, 2009).

In this work, we aim to circumvent these challenges. We propose a simple modification to model-free RL that allows the learned policy and value function to "look into the future" but without the need to learn an accurate model. Namely, we propose to modify the policy and value function to not only condition on the presently observed state and action but also on the subsequent trajectory (sampled by the agent as it was interacting with the environment) following this state and action. This way, our method mitigates potential approximation or feasibility issues in accurately modeling the future. To ensure that the learned policy and value function remains relevant during inference (i.e., data collection) when the future trajectory is unavailable, we place an *information bottleneck* (Saxe et al., 2018; Tishby et al., 1999) on the additional inputs, encouraging the learned functions to minimize their reliance on this privileged information. One may thus view our method as an instance of *teacher forcing* or *Z-forcing* (Goyal et al., 2017; Lamb et al., 2016) where our student is the learned policy and value function and the teacher is some function of the information in the future trajectory. It is well known that combining a strong autoregressive decoder with latent variables while ensuring that they carry useful information is difficult (Bowman et al., 2016; Chen et al., 2017). Z-forcing enforces the incorporation of relevant information by learning latent variables that predict the future steps.

Practically, our method, Policy Gradients Incorporating the Future (PGIF), is versatile and easy to implement. We use either a backwards RNN or a transformer to inject downstream information from the observed trajectories by way of latent variables, with a KL divergence regularization on these latents. We apply PGIF on top of a variety of off-the-shelf RL algorithms, including RNN-based PPO (Schulman et al., 2017), SAC (Haarnoja et al., 2018), and BRAC (Wu et al., 2020), and evaluate these algorithms on online and offline RL as well as sparse-reward and partially observable environments. In all of these domains, we demonstrate the ability of PGIF to achieve higher returns faster compared to these existing RL algorithms on their own, thus showing that our proposed method is both versatile and beneficial in practice.

## 2 BACKGROUND AND NOTATION

We begin by providing a brief overview of the notation and preliminary concepts that we will use in our later derivations.

**Markov Decision Processes (MDPs)** MDPs are defined by a tuple $\langle \mathcal{S}, \mathcal{A}, \mathbb{P}, R, \rho_0, \gamma \rangle$ where $\mathcal{S}$ is a set of *states*, $\mathcal{A}$ is a set of *actions*, $\mathbb{P}$ is a transition kernel giving a probability $\mathbb{P}(s'|s, a)$ over next states given the current state and action, $R : \mathcal{S} \times \mathcal{A} \to [R_{\min}, R_{\max}]$ is a reward function, $\rho_0$ is an initial state distribution, and $\gamma \in [0, 1)$ is a discount factor. An agent in this MDP is a stationary policy $\pi$ giving a probability $\pi(a|s)$ over actions at any state $s \in \mathcal{S}$. A policy $\pi$ interacts with the MDP by starting at $s_0 \sim \rho_0$ and then at time $t \geq 0$ sampling an action $a_t \sim \pi(s_t)$ at which point the MDP provides an immediate reward $R(s_t, a_t)$ and transitions to a next state $s_{t+1} \sim \mathbb{P}(s_t, a_t)$. The interaction ends when the agent encounters some terminal state $s_T$.

The value function $V^\pi : \mathcal{S} \to \mathbb{R}$ of a policy is defined as $V^\pi(s) = \mathbb{E}_\pi[\sum_{t=0}^{T-1} \gamma^t r_t | s_0 = s]$, where $\mathbb{E}_\pi$ denotes the expectation of following $\pi$ in the MDP and $T$ is a random variable denoting when a terminal state is reached. Similarly, the state-action value function $Q^\pi : \mathcal{S} \times \mathcal{A} \to \mathbb{R}$ is defined as $Q^\pi(s, a) = \mathbb{E}_\pi[\sum_{t=0}^{T-1} \gamma^t r_t | s_0 = s, a_0 = a]$. The advantage $A^\pi$ is then given by $A^\pi(s, a) = Q^\pi(s, a) - V^\pi(s)$. We denote $\rho_\pi$ as the distribution over trajectories $\tau = (s_0, a_0, r_0, \ldots, s_T)$ sampled by $\pi$ when interacting with the MDP.

During learning, $\pi$ is typically parameterized (e.g., by a neural network), and in this case, we use $\pi_\theta$ to denote this parameterized policy with learning parameters given by $\theta$. The policy gradient theorem (Sutton et al., 1999) states that, in order to optimize the RL objective $\mathbb{E}_{s_0 \sim \rho_0}[V^\pi(s_0)]$, a parameterized policy should be updated with respect to the gradient of the following loss (ignoring

any gradients through $\rho_{\pi_\theta}$, where we denote stopping the gradient with $\bar{\theta}$),

$$J_{\text{PG}}(\pi_\theta) = \mathbb{E}_{\tau \sim \rho_{\pi_{\bar{\theta}}}}[\textstyle\sum_{t=0}^{T-1} \gamma^t \cdot \hat{Q}_t \log \pi_\theta(a_t|s_t)], \quad (1)$$

where $\hat{Q}_t$ is an unbiased estimate of $Q^\pi(s_t, a_t)$. In the simplest case, $\hat{Q}_t$ is the empirically observed future discounted return following $s_t, a_t$. In other cases, an approximate $Q$-value or advantage function is incorporated to trade-off between the bias and variance in the policy gradients. When the $Q$ or $V$ value function is parameterized, we will use $\psi$ to denote its parameters. For example, the policy gradient loss with a parameterized $Q_\psi$ is given by,

$$J_{\text{PG}}(\pi_\theta, Q_\psi) = \mathbb{E}_{\tau \sim \rho_{\pi_\theta}} \left[ \sum_{t=0}^{T-1} \gamma^t \cdot Q_\psi(s_t, a_t) \log \pi_\theta(a_t|s_t) \right]. \quad (2)$$

The value function $Q_\psi$ is typically learned via some regression-based temporal differencing method. For example,

$$J_{\text{TD}}(Q_\psi) = \mathbb{E}_{\tau \sim \rho_{\pi_\theta}} \left[ \sum_{t=0}^{T-1} (\hat{Q}_t - Q_\psi(s_t, a_t))^2 \right]. \quad (3)$$

**Stochastic Latent Variable Models**   In our derivations, we will utilize parameterized policies and value functions conditioned on auxiliary inputs given by stochastic latent variables. That is, we consider a latent space $\mathcal{Z}$, typically a real-valued vector space. We define a parameterized policy that is conditioned on this latent variable as $\pi_\theta(a|s, z)$ for $a \in \mathcal{A}, s \in \mathcal{S}, z \in \mathcal{Z}$; i.e., $\pi_\theta$ takes in states and latent variables and produces a distribution over actions. In this way, one can consider the latent variable $z$ as modulating the behavior of $\pi_\theta$ in the MDP. During interactions with the MDP or during training, the latent variables themselves are generated by some stochastic process, thus determining the behavior of $\pi_\theta$. For example, in the simplest case $z$ may be sampled from a latent prior $p_{\upsilon^{(Z)}}(z|s)$, parameterized by $\upsilon^{(Z)}$. Thus, during interactions with the MDP actions $a_t$ are sampled as $a_t \sim \pi_\theta(s_t, z_t), z_t \sim p_{\upsilon^{(Z)}}(s_t)$. We treat parameterized latent variable value functions analogously. Specifically, we consider a latent space $\mathcal{U}$ and a parameterized $Q$-value function as $Q_\psi(s, a, u)$. A prior distribution over these latent variables may be denoted by $p_{\upsilon^{(U)}}(u|s)$.

## 3   POLICY GRADIENTS INCORPORATING THE FUTURE

Our method aims to allow a policy during training to leverage *future* information for learning control. We propose to utilize stochastic latent variable models to this end. Namely, we propose to train $\pi_\theta$ and $Q_\psi$ with latent variables $(\mathbf{z}, \mathbf{u}) = \{(z_t, u_t)\}_{t=0}^{T-1}$ sampled from a learned function $q_\phi(\tau)$ which has access to the full trajectory. For example, the PGIF form of the policy gradient objective in (2) may be expressed as

$$J_{\text{PGIF, PG}}(\pi_\theta, Q_\psi, q_\phi) = \mathbb{E}_{\tau \sim \rho_\pi, (\mathbf{z}, \mathbf{u}) \sim q_\phi(\tau)} \left[ \sum_{t=0}^{T-1} \gamma^t \cdot Q_\psi(s_t, a_t, u_t) \log \pi_\theta(a_t|s_t, z_t) \right]. \quad (4)$$

The PGIF form of the temporal difference objective in (3) may be expressed analogously as,

$$J_{\text{PGIF,TD}}(Q_\psi, q_\phi) = \mathbb{E}_{\tau \sim \rho_{\pi_\theta}, \mathbf{u} \sim q_\phi(\tau)} \left[ \sum_{t=0}^{T-1} (\hat{Q}_t - Q_\psi(s_t, a_t, u_t))^2 \right]. \quad (5)$$

It is clear that any RL objective which trains policies and/or value functions on trajectories can be adapted to a PGIF form in a straightforward manner. For example, in our experiments we will apply PGIF to an LSTM-based PPO (Schulman et al., 2017), SAC (Haarnoja et al., 2018), and BRAC (Wu et al., 2020).

While the PGIF-style objectives above adequately achieve our aim of allowing a policy to leverage future trajectory information during training, they also present a challenge during inference. When performing online interactions with the environment, one cannot evaluate $q_\phi(\tau)$, since the full trajectory $\tau$ is not yet observed. Therefore, while we want to give $\pi_\theta$ and $Q_\psi$ the ability to look at the full $\tau$ during training, we do not want their predictions to overly rely on this privileged information. To this end, we introduce a regularization on $q_\phi$ in terms of a KL divergence from a prior distribution $p_\upsilon(\tau) := \{p_\upsilon(z_t, u_t|s_t)\}_{t=0}^{T-1}$ which conditions $(z_t, u_t)$ only on $s_t$. Thus, in the case of policy gradient, the full loss is,

$$J_{\text{PGIF-KL}}(\pi_\theta, Q_\psi, q_\phi) = J_{\text{PGIF, PG}}(\pi_\theta, Q_\psi, q_\phi) + \beta \mathbb{E}_{\tau \sim \rho_\pi} \left[ D_{\text{KL}}(q_\phi(\tau) \| p_\upsilon(\tau)) \right], \quad (6)$$

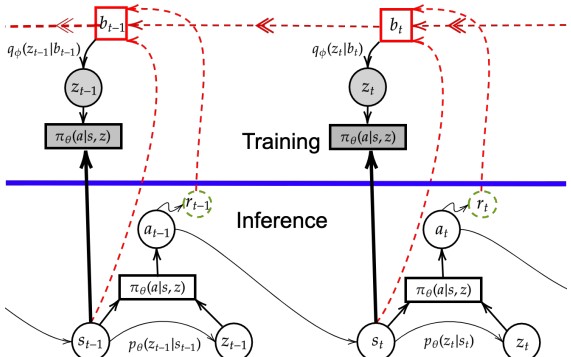

Figure 1: The architecture of the model. Our inference model $q_{\phi^{(z)}}$ uses a backwards hidden state $b_t$ to approximate dependencies of $z_t$ on the future of the trajectory. The blue line separates the data collection and policy gradient training steps in our algorithm and the red lines represent information flowing into the backwards RNN. Grey variables are used during training and white variables are used during data collection. **Top:** we show the training model where the policy gradient loss is calculated with backwards RNN hidden state information. **Bottom:** we show the data collection phase of the algorithm utilizing latent variables sampled from the latent prior.

where $\beta$ is the weight of the divergence term. The introduction of this prior thus solves two problems: (1) it encourages the learned policies and value functions to not overly rely on information beyond the immediate state; (2) it provides a mechanism for inference, namely using latent samples from $p_\upsilon(s)$ when interacting with the environment.

**Parameterization of** $q_\phi$    In our implementation, we parameterize $q_\phi(\tau)$ as an RNN operating in reverse order on $\tau$. Specifically, we use an LSTM network to process the states in $\tau$ backwards, to yield LSTM hidden states $\mathbf{b} = \{b_t\}_{t=0}^{T-1}$. The function $q_\phi$ is then given by Gaussian distributions with mean and variance at time $t$ derived from the backwards state $b_t$. In practice, to avoid potential interfering gradients from the objectives of $\pi_\theta$ and $Q_\psi$, we use separate RNNs with independent parameters $q_{\phi^{(z)}}, q_{\phi^{(u)}}$ for $z_t, u_t$, respectively. See Figure 1 for a graphical diagram of the training and inference procedure for PGIF. In our empirical studies shown in Appendix G, we will also show that a transformer (Vaswani et al., 2017) can be used in place of an RNN with minimal decrease in performance, providing more computational efficiency and potentially allowing for better propagation of information over time.

### 3.1 Variational Information Bottleneck Interpretation

The KL regularization we employ above may be interpreted as a variational information bottleneck, constraining the mutual information between the latent variable distribution and the trajectory $\tau$. Here we provide a brief derivation establishing this equivalence.

For simplicity, we consider a specific timestep $t \in \mathbb{N}$ and a starting state $s_t = s$. Let $\mathcal{T}_{\geq t}$ denote the random variable for all information contained after and including timestep $t$ in trajectory $\tau$ induced by $\pi$. Let $\mathcal{U}_t$ be the random variable for latents $u_t$ induced by $q_\phi(\tau_{\geq t})$ conditioned on all steps in the trajectory after and including $t$. Consider a constrained objective minimizing $J_{\text{PGIF,TD}}$ while enforcing an upper bound $I_{\max}$ on the mutual information between the distribution of trajectory steps and the distribution of latent variables $I(\mathcal{T}_{\geq t}, \mathcal{U}_t | s_t = s)$. This objective is given by,

$$\min_{\psi, \phi} \quad J_{\text{PGIF,TD}}(Q_\psi, q_\phi | s_t = s) := \mathbb{E}_{\tau \sim \rho_{\pi_\theta}(\cdot | s_t = s), u_t \sim q_\phi(\tau_{\geq t})}[(\hat{Q}_t - Q_\psi(s_t, a_t, u_t))^2] \tag{7}$$

$$\text{s.t. } I(\mathcal{T}_{\geq t}, \mathcal{U}_t | s_t = s) \leq I_{\max}.$$

Recall the definition of mutual information:

$$I(\mathcal{T}_{\geq t}, \mathcal{U}_t | s_t = s) = \int p(\tau_{\geq t}, u_t | s_t = s) \log \frac{p(\tau_{\geq t}, u_t | s_t = s)}{p(\tau_{\geq t} | s_t = s) p(u_t | s_t = s)} du_t d\tau_{\geq t}$$

$$= \int \rho_{\pi_\theta}(\tau_{\geq t} | s_t = s) q_\phi(u_t | \tau_{\geq t}) \log \frac{q_\phi(u_t | \tau_{\geq t})}{p(u_t | s_t = s)} du_t d\tau_{\geq t}, \tag{8}$$

where $p(u_t|s_t = s)$ is the marginal distribution over the latent variable $p(u_t|s_t = s) = \int q_\phi(u_t|\tau_{\geq t})\rho_\pi(\tau_{\geq t}|s_t = s)d\tau_{\geq t}$. This marginal is intractable to compute directly, and so to approximate this marginal we introduce a variational distribution $h(u_t|s_t = s)$. By definition we know that $D_{\text{KL}}[p(u_t|s_t = s)\|h(u_t|s_t = s)] \geq 0$. We can then see that $\int p(u_t|s_t = s)\log p(u_t|s_t = s)du_t \geq \int p(u_t|s_t = s)\log h(u_t|s_t = s)du_t$. We therefore derive the upper bound for use in equation 7 as,

$$I(\mathcal{T}_{\geq t}, \mathcal{U}_t|s_t = s) \leq \int \rho_\pi(\tau_{\geq t}|s_t = s)q_\phi(u_t|\tau_{\geq t})\log \frac{q_\phi(u_t|\tau_{\geq t})}{h(u_t|s_t = s)}du_t d\tau_{\geq t}$$
$$\leq \mathbb{E}_{\tau \sim \rho_{\pi_\theta}(\cdot|s_t = s)}\Big[D_{\text{KL}}(q_\phi(u|\tau_{\geq t})\|h(u_t|s_t = s))\Big]. \tag{9}$$

We can subsume the constraint into the objective as,

$$\min_{\psi,\phi} \quad J_{\text{PGIF,TD}}(Q_\psi, q_\phi|s_t = s) + \beta\left(\mathbb{E}_{\tau \sim \rho_{\pi_\theta}(\cdot|s_t = s)}[D_{\text{KL}}(q_\phi(\tau_{\geq t})\|h(u_t|s_t = s))] - I_{\text{max}}\right).$$

By taking $h$ to be our learned prior $p_v$, we recover the single step ($s_t = s$), TD analogue of the PGIF objective in (6), offset by a constant $\beta \cdot I_{\text{max}}$, as desired.

## 3.2 Z-Forcing with Auxiliary Losses

While our proposed training architecture enables the policy and value function to look at the full trajectory $\tau$, in practice it may be difficult for the trajectory information to propagate, especially in settings with highly sparse learning signals. In fact, it is known that such latent variable models may ignore the latent variables due to optimization issues, completely negating any potential benefit (Bengio et al., 2015). To circumvent these issues, we make use of the idea of *Z-forcing* (Goyal et al., 2021), which employs auxiliary losses and models to force the latent variables to encode information about the future. We denote this loss as $J_{\text{Ax}}(\zeta)$ where $\zeta$ is the set of parameters in any auxiliary models, and elaborate on the main forms of this loss which we consider below. We do not combine different types of auxiliary losses in our total loss function.

We emphasize that these auxiliary losses are applied exclusively to the backwards encoder $q_\phi(\tau)$ and have no direct impact on the learned policy or value functions. In fact, we conduct ablations in Appendix F showing that these auxiliary losses applied directly to policy and value functions without PGIF perform worse.

**State based forcing (Force)**    A simple way to force state information to be encoded is to derive conditional generative models $p_\zeta(b_t|z_t)$ over the backwards states given the inferred latent variables $z_t \sim q_{\phi^{(Z)}}(z_t|b_t)$, and similarly for the latents $u_t$. We can write this auxiliary objective as a maximum log-likelihood loss $J_{\text{Ax}}(\zeta) = -\mathbb{E}_{q_{\phi^{(Z)}}(z_t|b_t)}[\log p_\zeta(b_t|z_t)]$. This way, we enforce the noisy mapping $b_t \to z_t$ defined by $q_{\phi^{(Z)}}$ to not be *too noisy* so as to completely remove any information from $b_t$.

**Value prediction networks (VPN)**    A more sophisticated approach to force information to be propagated is to use an autoencoder-like, model-based auxiliary loss. To this end, we take inspiration from VPNs (Oh et al., 2017), and apply an auxiliary loss that uses $b_t$ to predict future rewards, values, and discounts. Note that, in principle, $b_t$ already has access to this information, by virtue of the backwards RNN or transformer conditioned on the future trajectory. Thus, this auxiliary loss only serves to enforce that the RNN or transformer dutifully propagates this information from its inputs. We also note that, in contrast to the state based forcing described above, this approach only enforces $b_t$ to contain the relevant information, and it is up to the RL loss whether this information should be propagated to the latents $z_t, u_t$. We give a detailed explanation of VPNs in Appendix B.

## 3.3 Full Algorithm

The full learning objective for PGIF is thus composed of three components: First, a latent-variable augmented RL objective, e.g., policy gradient as shown in (4). Second, a KL regularizer, e.g., as shown in (6). Finally, an auxiliary loss, given by either state based forcing (Force) or value prediction networks (VPN). We present an example pseudocode of a PGIF-style policy gradient with learned value function in Algorithm 1. See Appendix C for further details, including how to adaptively tune the coefficients on the KL and auxiliary loss components as well as more specific pseudocode algorithms for advantage policy gradient and soft actor-critic.

---

**Algorithm 1** PGIF Algorithm with State-Action Value Function Estimation

---

**Require:** Initial parameters: $\theta, \upsilon^{(U)}, \upsilon^{(Z)}, \phi^{(U)}, \phi^{(Z)}, \psi, \zeta_{PG}, \zeta_{TD}$, Weights: $\alpha_{PG}, \alpha_{TD}, \beta_{PG}, \beta_{TD}$
 1: **for** policy-step $k = 0, 1, 2, \ldots, N$ **do**
 2:     Collect set of Trajectories $\mathcal{D} = \{\tau_i, \ldots\}$ :
 3:     **repeat**
 4:         $z_t \sim p_{\upsilon^{(Z)}}(z_t)$
 5:         Execute: $\pi_\theta(a_t|s_t, z_t)$ and observe $r_t, s_{t+1}$ from environment.
 6:     **until** episode termination
 7:     **for** Trajectory: $\tau_i \in \mathcal{D}$ **do**
 8:         $\mathbf{b^Z} = \text{BackwardsLSTM}^Z(\tau_i)$ (Operates over the entire trajectory)
 9:         $\mathbf{b^U} = \text{BackwardsLSTM}^U(\tau_i)$
10:         $\tau_i = \tau_i \cup \{\mathbf{b^Z}, \mathbf{b^U}\}$
11:     $\forall \{\mathbf{s}, \mathbf{a}, \mathbf{r}, \mathbf{b^Z}, \mathbf{b^U}\} \in \mathcal{D}$:
12:     Derive $J_{\text{Ax-PG}}(\zeta_{PG}), J_{\text{Ax-TD}}(\zeta_{TD})$ according to any auxiliary loss.
13:     $D_{TD} = D_{KL}(q_{\phi^{(U)}}(\mathbf{u}|\mathbf{b^U})\|p_{\upsilon^{(U)}}(\mathbf{u}|\mathbf{s}))$
14:     $J_{TD} = \mathbb{E}_{\tau \in \mathcal{D}}\left[J_{\text{PGIF,TD}}(Q_\psi, q_{\phi^{(U)}}) + \alpha_{TD}J_{\text{Ax-TD}}(\zeta_{TD}) + \beta_{TD}D_{TD}\right]$
15:     $D_{PG} = D_{KL}(q_{\phi^{(Z)}}(\mathbf{z}|\mathbf{b^Z})\|p_{\upsilon^{(Z)}}(\mathbf{z}|\mathbf{s}))$
16:     $J_{PG} = \mathbb{E}_{\tau \in \mathcal{D}}\left[J_{\text{PGIF, PG}}(\pi_\theta, Q_\psi, q_{\phi^{(Z)}}) + \alpha_{PG}J_{\text{Ax-PG}}(\zeta_{PG}) + \beta_{PG}D_{PG}\right]$
17:     Update all parameters w.r.t: $J_{PG}$ and $J_{TD}$

---

## 4 RELATED WORK

We review relevant works in the literature in this section, with additional discussions in Appendix B.

**Incorporating the future**   Recent works in model-based RL have considered incorporating the future by way of dynamically leveraging rollouts of various horizon lengths and then using them for policy improvement (Buckman et al., 2018) . Z-forcing and stochastic dynamics models have been applied directly to learning environmental models and for behavioral cloning while incorporating the future but not for online or offline continuous control (Ke et al., 2019). Our present work is unique for incorporating Z-forcing and conditioning on the future in the model-free RL setting. A few other methods explore the future in less direct ways. For example, RL Upside down (Schmidhuber, 2020) uses both reward (or desired return) and state to predict actions, turning RL into a supervised learning problem.

**Hindsight**   Hindsight credit assignment introduces the notion of incorporating the future of a trajectory by assigning credit based on the likelihood of an action leading to an outcome in the future (Harutyunyan et al., 2019). These methods were extended using a framework similar to ours, leveraging a backwards RNN to incorporate information in hindsight (Mesnard et al., 2021). Still, there are a number of differences compared to our own work. (1) Only the value function (rather than both the value and policy functions) is provided access to the future trajectory, whereas we show that allowing the actor access has benefits in some tasks (Appendix F). (2) There is no KL information bottleneck; rather information is constrained via an action prediction objective. (3) These previous works do not employ any Z-forcing, while it is well-known that learning useful latent variables in the presence of an autoregressive decoder is difficult without Z-forcing (Bayer & Osendorfer, 2015); in fact, in our own preliminary experiments we found our algorithm performs significantly worse without any auxiliary losses. Value driven hindsight modelling (HiMo) (Guez et al., 2020) proposes a hindsight value function, which is conditioned on future information in the trajectory. In this work, they primarily use the hindsight value function (separate from the agent's value function) to learn a low-dimensional representation, which is distilled to a non-hindsight representation that is then used by the actual actor and critic. Thus there are a few key differences from our work: (1) There are no gradients passing from the RL loss to the representation loss (only the separate value prediction loss is used to train the representation), thus this method is arguably less end-to-end than PGIF; (2) The only mechanism for controlling the amount of information in the representation is through its dimension size and look-ahead, while using a KL penalty is more flexible. Our method is more versatile than previous works, being applicable to off-policy and offline RL settings rather than purely on-policy RL, as in these previous works. Nevertheless, it is an interesting avenue for future work to investigate how to combine the best of both approaches, especially with guarantees on variance reduction of policy gradient estimators (Nota et al., 2021) and hierarchical policies (Wulfmeier et al., 2021).

## 5    EXPERIMENTS

We now provide a wide array of empirical evaluations of our method, PGIF, encompassing tasks with delayed rewards, sparse rewards, online access to the environment, offline access to the environment, and partial observability. In the appendix, we include further demonstrations of PGIF applied to the challenging AntMaze environment (Sec. H) with substantial performance improvements, online RL with full observability (Sec. E) with improvements over a SAC baseline, and numerous ablation analyses (Sec. F, I) identifying the components of PGIF that are responsible for performance.

In our online RL experiments, we compare against Soft Actor Critic (SAC) (Haarnoja et al., 2018) and Proximal Policy Optimization (PPO) (Schulman et al., 2017) with an LSTM layer in the policy network. The comparison with PPO is particularly important since this method also leverages a form of artificial memory of the past (but not forward-looking like in PGIF). SAC is a state-of-the-art model-free RL method that employs a policy entropy term in a policy gradient objective and shows optimal performance and stability when compared with other online deep RL benchmarks. We show the hyper-parameters for each experiment in the Appendix D. In addition, we explore using our method with a transformer as opposed to an LSTM as the backwards network in Appendix G.

### 5.1    CREDIT ASSIGNMENT AND SPARSE REWARDS

PGIF aims to provide a better representation for credit assignment by accounting for downstream information rather than current. This can be used as additional input to the value function or policy. We find this helps address two very related problems, optimal credit assignment and learning with sparse rewards. The former problem is exasperated by the latter since rewards are rarely obtained, possibly only at the end of the episode. The relevance of each action (*credit*) must be assigned using only these rare rewards.

| Method | Mean Bsuite-score |
|---|---|
| PPO-LSTM | $0.33 \pm 0.09$ |
| SAC | $0.41 \pm 0.03$ |
| DQN (3-step return) | $0.44 \pm 0.06$ |
| DQN (1-step return) | $0.51 \pm 0.03$ |
| PGIF-PPO (VPN) | $0.46 \pm 0.09$ |
| PGIF-PPO (Force) | $0.26 \pm 0.10$ |
| PGIF-SAC (VPN) | $\mathbf{0.60} \pm 0.04$ |
| PGIF-SAC (Force) | $\mathbf{0.58} \pm 0.08$ |
| PGIF-DQN (Force) | $0.58 \pm 0.04$ |

Table 1: Performance on the Umbrella-Length environment. We run our model for 1000 episodes steps over 5 random seeds. The BSuite score is calculated in terms of the regret normalized $[\text{random}, \text{optimal}] \rightarrow [0, 1]$ (higher is better). The number after $\pm$ is the standard deviation.

We first aim to show that our method is effective in a simple environment where credit assignment is the paramount objective. We examine the Umbrella-Length task from BSUITE (Osband et al., 2020), a task involving a long sequential episode where only the first observation (a forecast of rain or shine) and action (whether to take an umbrella) matter, while the rest of the sequence contains random unrelated information. A reward of $+1$ is given at the end of the episode if the agent chooses correctly to take the umbrella or not depending on the forecast. This difficult task is used to test the agent's ability to assign credit correctly to the first decision. We evaluate PGIF-versions of SAC and PPO to vanilla discrete-SAC (Haarnoja et al., 2018; Christodoulou, 2019) as well as a PPO-LSTM (Schulman et al., 2017) baseline. Results are presented in Table 1. We see that our method is able to achieve best performance. This suggests that the PGIF agent is able to efficiently and accurately propogate information about the final reward to the initial timestep, more so than either the one-step backups used in SAC or the multi-step return regressions used in PPO-LSTM can.

We continue to the Gym-MiniGrid (Chevalier-Boisvert et al., 2018) set of partially-observable environments, for which rewards are sparse and there is a non-zero reward only when the agent completes the task (with the reward proportional to the time taken). The agent is only given a local ego-centric view of its environment. For example, in *DoorKey* a key must first be obtained and then a door opened leading to another room with the goal state. It is difficult, but essential, to assign credit to obtaining the key and opening the door in the correct sequence. We compare against PPO-LSTM and an implementation of HiMo, which also suggests to use information from the future during agent learning (Guez et al., 2020). We show the results of these experiments and examples of the grids in Figure 2, with PGIF-PPO outperforming both baselines. The difference is especially stark on the *Unlock* Environment where credit assignment is the key challenge and the standard PPO baseline

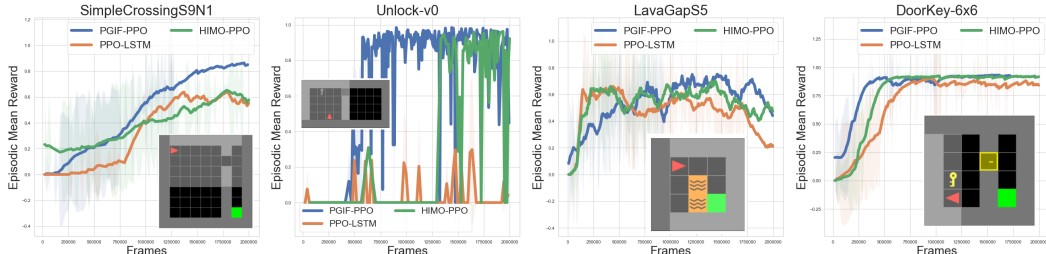

Figure 2: The online episodic mean reward evaluated over 5 episodes every 250 steps for MiniGrid RL tasks. We show the average over 5 random seeds. $2M$ environment step interactions are used. The shaded area shows the standard error. PGIF always uses state based forcing in these environments.

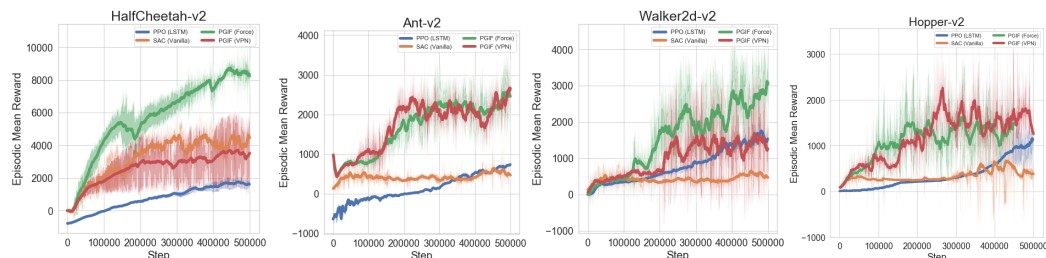

Figure 3: The online episodic mean reward evaluated over 5 episodes every 500 steps for MuJoCo continuous control RL tasks with partial observability. We show the average over 5 random seeds. $500,000$ environment step interactions are used. The shaded area shows the standard error.

fails to converge to any rewards. In LavaGap, there is no significant improvement. We hypothesize that this is because the most common trajectories generated during early training are ones where the agent enters the lava. There is also no negative reward for entering the lava, and the episode is simply terminated, making it more difficult to avoid. This creates a situation where our future representation is highly suboptimal, especially during early training.

## 5.2 PARTIAL OBSERVABILITY

We now aim to show that our method is not only effective in fully-observed Markovian settings, but also in environments with partial observability. This set of experiments uses the MuJoCo robotics simulator (Todorov et al., 2012) suite of continuous control tasks. These are a set of popular environments used in both online and offline deep RL works (Fujimoto et al., 2019; 2018) and provides an easily comparable benchmark for evaluating algorithm sample efficiency and reward performance. As in previous work (Yang & Nachum, 2021), we introduce an easy modification to these tasks to make the environment partially observable thereby increasing the difficulty: We zero-out a random dimension of the state space at each data collection step. This helps us test partial observability, a feature common in robotic agent tasks.

We compare a PGIF-style SAC implementation to vanilla SAC and PPO-LSTM on these domains. We show the results of these experiments in Fig. 3. We find that PGIF can provide improved performance on these difficult tasks, suggesting that PGIF is able to leverage future information in the trajectory to appropriately avoid uncertainties about the environment, more so than when only conditioning on the immediate state (vanilla SAC) or even when conditioning on the entire past trajectory (PPO-LSTM). Interestingly, we find that the simple state based forcing (Force) performs more consistently better than the more sophisticated VPN based forcing. See Appendix E for online evaluations without partial observability.

## 5.3 OFFLINE RL EVALUATIONS

To asses if our method is effective in an offline RL setting, we evaluate our proposed algorithm in several continuous control offline RL tasks (Fu et al., 2020) against Behavior Regularized Actor

Critic (BRAC) (Wu et al., 2020) and Batch-Constrained Q-learning (BCQ) (Fujimoto et al., 2019). BRAC operates as a straightforward modification of SAC, penalizing the value function using a measure of divergence (KL) between the behaviour policy and the learned agent policy. For our PGIF algorithm, we use BRAC as the starting point. For these offline MuJoCo tasks, we examine D4RL datasets classified as *medium* (where the training of the agent is ended after achieving a "medium" level performance) and *medium expert* (where medium and expert data is mixed) (Fu et al., 2020). Datasets that contain these sub-optimal trajectories present a realistic problem for offline RL algorithms. We also include an offline version of the AntMaze, which is particularly challenging due to sparse rewards. We show the results of these experiments in Table 2. We find that our method outperforms the baselines in all but one of the tasks in terms of final episodic reward. We hypothesize in the medium-expert setting that we perform slightly worse due to the lack of action diversity which makes learning a dynamics representation difficult. Interestingly, in contrast to the online results, we find that VPN based forcing performs better than state based forcing, although even state based forcing usually performs better than the baseline methods.

| Environment | BRAC | PGIF (VPN) | PGIF (Force) | BCQ |
|---|---|---|---|---|
| ant-medium | $2731 \pm 329$ | $\mathbf{3250} \pm 125$ | $2980 \pm 164$ | $1851 \pm 94$ |
| ant-medium-expert | $2483 \pm 329$ | $\mathbf{3048} \pm 362$ | $2431 \pm 417$ | $2010 \pm 133$ |
| hopper-medium | $1757 \pm 183$ | $\mathbf{2327} \pm 399$ | $1930 \pm 44$ | $1722 \pm 166$ |
| walker2d-medium | $3687 \pm 25$ | $\mathbf{3989} \pm 259$ | $3821 \pm 341$ | $2653 \pm 301$ |
| halfcheetah-medium | $5462 \pm 198$ | $6037 \pm 324$ | $\mathbf{6231} \pm 303$ | $4722 \pm 206$ |
| halfcheetah-medium-expert | $\mathbf{5580} \pm 105$ | $5418 \pm 76$ | $5491 \pm 143$ | $4463 \pm 88$ |
| antmaze-umaze | $0.5 \pm 0.16$ | $\mathbf{0.95} \pm 0.0$ | $0.7 \pm 0.15$ | $0.8 \pm 0.13$ |

Table 2: Performance on the offline RL tasks showing the average episodic return. The final average return is shown after training the algorithm for $500,000$ episodes and then evaluating the policy over $5$ episodes. Results show an average of $5$ random seeds. The value after $\pm$ shows the standard error.

## 6 DISCUSSION

In this work, we consider the problem of incorporating information from the entire trajectory in model-free online and offline RL algorithms, enabling an agent to use information about the future to accelerate and improve its learning. Our empirical results attest to the versatility of our method. The benefits of our method are apparent in both online and offline settings, which is a rare phenomenon given that many previous offline RL works suggest that what works well in online RL often transfers poorly to offline settings, and vice versa (Fujimoto et al., 2019). Beyond just online and offline RL, our results encompass partial observability, sparse rewards, delayed rewards, and sub-optimal datasets, demonstrating the ability for PGIF to achieve higher reward faster in all settings.

We also wish to highlight potential risks in our work. Specifically, the use of future information in PGIF may exacerbate biases present in the experience or offline dataset. For example, it is known that expert datasets lack action diversity. Further conditioning on the future in this dataset could force these biases to be incorporated more easily into the learning algorithm. Some other biases that may arise are a trajectory that contains some marginal reward and is therefore incorporated into our policy/value function with PGIF. This could hamper exploration of the agent and prevent discovery of states that yield optimal reward. It may be interest to combine our method with exploration strategies. Perhaps there are benefits in decreasing access to future information during initial exploratory agent steps so the agent is better able to explore. Furthermore, we find that our method is slower to train than the baselines we compare to, due to the fact that the architecture requires training an LSTM, with at times, very long trajectories as input. We must use the entire trajectories to train this LSTM. These issues can largely be alleviated by using a transformer, with minimal difference in performance.

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
