# OpenReview forum: "Policy Gradients Incorporating the Future"
_ICLR.cc/2022/Conference — ICLR 2022 Poster_

### Official Review · Reviewer_GXWR · 2021-10-20

**Correctness:** 3
**Technical Novelty And Significance:** 3
**Empirical Novelty And Significance:** 3
**Recommendation:** 6
**Confidence:** 3

**Main Review:**

Strengths
------------------------------------
- The clarity was mostly good.
- Experimental results appeared good (in terms of improving over the baselines, but I am not sure the baselines were well-implemented).
- The method is general and could be applied together with most RL algorithms.
- The method improved performance for both online and offline RL algorithms.

Weaknesses
------------------------------------
- It wasn't clear to me why the method should improve the performance.
- The experiments only looked at reward scores without examining any other statistic or property of the learning.
- The computational time is much larger compared to other methods (they said it's much slower, but unfortunately did not say exactly by how much).
- It was not clear to me that the baseline algorithms were solid, as it seems that the experimental results are new, and not reproductions of previous results in other publications.
- Some ablation studies are missing (in particular, adding a prior network to the models essentially increases the capacity of the Q and policy models, so I thought it should be checked whether the baseline algorithms may also benefit from using larger models).

Suggested decision
------------------------------------
Currently, I am leaning toward recommending to reject, as I cannot recommend accepting a paper unless I am sure the empirical results are solid. Potentially, this can be cleared up in the rebuttal.

Supporting arguments
------------------------------------
**Experiments**

I can't think of any good theoretical reason why one would want to do what the authors did in their paper, so I think that the experimental results are crucial to convey usefulness and validity of their approach. Currently, I was not convinced that the experiments are solid (see below), and I also think that reporting only reward scores is quite a limited method of experimentation as it does not explain why the performance may have improved. One of the clearest ways to show results that can be trusted is to reproduce existing results from another publication, then outperform those results, but it didn't seem like this was done.

The Bsuite umbrella-length experiment seems fine. The result of ~0.6 is higher than the result of ~0.5 for DQN reported in the bsuite paper. However, the new baselines in this paper (PPO and SAC) only had a score of ~0.4. Also, why were none of the other bsuite environments tested?

In minigrid there is also no reference to any previous publication showing the same results as produced in this paper.

The experiments with partial observability were based on the work of Yang & Nachum, (2021), but it appears executed in a different way, and it's not easy to see that the baselines reproduced any results from their work.

The offline RL experimental results also are not clear to me. For example, on halfcheetah your baselines achieve around 4.5k score, but in the BRAC paper, the algorithms achieve around 6k score. Where does the discrepancy come from?

Also, in the antmaze results in Table 10, your baselines achieve ~0, whereas the in the original paper
by Florensa et al. the baseline algorithms reached around 0.6 reward.

It is quite possible that the experiments are performed solidly; however, this is not explained sufficiently for me to understand this. I would appreciate if the authors can comment on the correctness of their implementations, and provide evidence that the results achieved by their baseline algorithms are competitive (e.g., by providing references to papers where the scores are similar to the baselines implemented in this paper).

**Concept of algorithm**

The algorithm was motivated from the point of view of allowing to "look into the future" to improve the optimization performance; however, typical policy gradient algorithms already do this when using Monte Carlo returns. Instead of using your approach, one can give perfect information of the future by just using the empirical return. Then what is the motivation of the method, and why should it help with learning?  One could think of some possible reasons such as variance reduction (perhaps it is doing something similar to variance elimination https://arxiv.org/abs/1811.06225 or http://proceedings.mlr.press/v100/cheng20a.html), or better representation learning; however, such points are not explored.

**Ablation studies**

The prior network $p(z|s)$ is essentially increasing the capacity of the policy network. I think it would be better to also compare to a version where the backward RNNs are not trained, but the $p(z|s)$ is just trained end-to-end as if it were a part of the policy with some additional latent noise $z$. It would also be interesting to know what happens if you increase the number of parameters of the critic (to account for the greater model capacity achieved by using the backward RNN).

Other notes
------------------------------------

Is the prior network also trained using the policy gradient loss (the action depends on the sampled $z$ from the prior, so it could be trained end-to-end)?

The discussion around equations 1 and 2 is not wrong, but
I think it would be better to explain that this is a surrogate loss
that allows easily implementing the policy gradient using
automatic differentiation, and that only the $\log \pi$ term will
be differentiated. Originally, the policy gradient theorem was not
presented in the way that you did.

On page 5 in the VPN section, what do you mean by predicting the discounts?
Isn't the discount factor just a constant? Also, is the encoding network
trained end-to-end?

"Our method is more versatile that previous works,"
-> "...versatile than..."

The figure axis labels are small and hard to read. Also, you used a
non-vector figure format, which makes them blurry. I prefer vector
format figures so that they look sharp when you zoom in. Moreover,
it would be better to use more than color to differentiate between
the lines.

"Furthermore, we find that our method is much slower to train than the
baselines we compare to, due to the fact that the architecture
requires training an LSTM, with at times, very long trajectories as
input."
It is good that you mentioned this, but can you roughly quantify it?
How much slower was it?

Table 7 top right, are you missing one digit in the PGIF score?

**Summary Of The Paper:**

The work proposes to run two recurrent LSTM neural networks backwards from the end of an episode, giving the rewards $r$ and states $s$ as input to the networks, and producing two separate latent state ($z$, $u$) distributions for each time step. Such procedure can be done during training after having collected an episode. Once, the $z$ and $u$ are computed, these are given as an input to the Q-function $Q(s,a,u)$, and the policy network $\pi(a|s,z)$ (note that the action was already sampled during the episode, but here the probability is recomputed using the new $z$). And the policy gradient/critic updates are compute as usual together with this augmented state. One remaining question is how to pick the $z$ during the episode when the future states are not known yet. This is done by training a prior network $p(z|s)$ by minimizing the KL divergence to the latent state distribution obtained from the backwards network (then one can just sample a $z$ from this prior network during the episode). To aid the backwards network in learning meaningful latent representations, they use z-forcing, which is a technique where the variable $z$ is used directly to predict some quantity to ensure that it incorporates useful information. They consider two options: 1) predict $b$ the hidden state of the backward RNN, 2) predict the values, rewards, discounts.
(Note that this z-forcing is done for both $z$ and $u$).

They perform experiments on Bsuite umbrella-length, Gym-minigrid, custom partially observable MuJoCo environments, Offline RL on MuJoCo D4RL tasks. They compare with PPO, SAC, and other baselines. The experimental results improved in all cases, but their method also requires more computational time due to having to train the LSTMs.

**Summary Of The Review:**

The experimental results look impressive, but I was not convinced that the baselines were well-implemented. The experimentation also only looked at reward scores, so it is not clear why the performance improved. It was also not clear to me that there is any good theoretical reason why the method should improve the performance. Currently, I am recommending to reject. I hope the authors will provide strong arguments and evidence in their rebuttal.

---------------------------------------------------------------------------------------------------------------------
Update after author rebuttal
----------------------------------------
The authors have provided additional evidence on the correctness of their baselines.
They also performed additional ablations, such as checking the change in performance when increasing
the model capacity. I was satisfied with these changes, so I have increased the correctness to 3, and the score the 6.
My recommendation is a borderline accept, as I was satisfied that the method empirically improves the performance; however,
I think the authors could have done more to examine why their method improves the performance.

I still have some concerns with the paper:
- The experiments still only look at the reward curves. They could have explored points such as the prediction accuracy of the value function, the variance and accuracy of the gradients, or tried to do some experiment to examine the quality of the representations (although this one is more difficult, as I don't think there is a standard way to look into this).
- I am still not convinced by the authors' explanation of why "looking into the future" is useful. When the information bottleneck is removed, the value function would ideally just predict the Monte Carlo return. So the authors claim appears to be that "looking into the future" is good, but if it is done too much, then it stops being useful. I think this point could have been examined in more detail, e.g. by doing a study on how the information bottleneck coefficient affects the performance (admittedly, it would have been better for me to say this in my initial review). Previous works (e.g., https://arxiv.org/abs/2011.09464) doing related ideas provided longer more principled explanations of why their methods would be useful by arguing that it might reduce the policy gradient variance by removing the random fluctuations in the future.

Some more minor concerns:
- I still think the discount prediction requires more clarification in the appendix B.
- Regarding the discussion around equations 1 and 2. I think you still
need to give more clarification. If one differentiates this "loss"
twice, one does not obtain the correct Hessian of the RL objective
(e.g., https://arxiv.org/abs/1802.05098).  I think this section should
be explained as an automatic differentiation trick to obtain first
order policy gradients.

---

> ### Author Response · Authors · 2021-11-19
> **Reviewer GXWR Response (1)**
>
> Thanks for closely reading our submission! We address your feedback below. Please let us know if this addresses all your concerns, and if so, whether you will reconsider your overall score.
>
> We would like to emphasize the reproducibility of our work.  We compile a list of our experiments and give examples of publications with nearly identical results. We then highlight some of our main rebuttal points.
>
>
> * For the Bsuite experiment in Table 1, we have now reproduced the DQN baseline and achieved a result close to 0.5, as in the original paper (Osband, 2019: https://arxiv.org/abs/1908.03568). We also show results of PGIF-DQN, exhibiting a significant improvement.
> * For MiniGrid sparse reward experiments in Figure 2, there are very few publications using the credit assignment environments that we utilized.  There does exist a paper using DQN (Co-Reyes, 2021: https://arxiv.org/pdf/2101.03958.pdf) and for environments like DoorKey and Unlock, the reward is very low with the standard DQN baseline.
> * For our offline RL experiments in Table 2 we use the D4RL set of environments and offline datasets.  We have reproduced the results on BRAC and BCQ nearly identically. D4RL (Fu, 2020: https://arxiv.org/abs/2004.07219).
> * For the online Soft Actor Critic experiments in Figure 4, we examined the original soft actor critic paper (Haarnoja, 2018: https://arxiv.org/abs/1801.01290) and found that we achieve extremely similar reward curves for the first 500k steps (in Appendix Section E in our work and Figure 1 in the SAC paper).
> * For the AntMaze results in Table 10, it is clear from other publications (Nachum, 2018: https://arxiv.org/abs/1805.08296, Nachum, 2019: https://arxiv.org/pdf/1810.01257.pdf, Li, 2019: https://arxiv.org/pdf/1910.04450.pdf) that the domain-agnostic baseline score is in fact 0.
> * For the partially observable online MuJoCo environments in Figure 3, we followed a similar procedure to Yang and Nachum, 2021 by randomly zero masking a state.  We give a detailed description below on how we have reproduced these results and how to interpret the results from this work.  For Hopper and Walker2d, the performance of SAC from Yang/Nachum is close to our own baseline SAC performance.  For Ant, there is a large discrepancy, and this is due to the fact that we zero out an extra dimension since Ant has a much larger state dimension than the other MuJoCo environments. We did this to make the problem slightly more difficult so we could better showcase our improvements (it is clear from the Yang/Nachum results that zeroing out a single dimension only negligibly affects the difficulty).  For HalfCheetah, our reward is slightly higher than the reported baseline. HalfCheetah can have high variance results depending on the network initialization.  We also found that our results on HalfCheetah were high variance (the standard error across all seeds, taking the average value over all training iterations is ~1532 for baseline SAC), with some seeds generating rewards corresponding to those in the Yang/Nachum paper. We believe this is the cause of the discrepancy in HalfCheetah baseline numbers.
> * As for the motivation of this work, we were inspired by using Z-forcing for learning models that incorporate the long term future (Ke, 2019: https://arxiv.org/abs/1903.01599).  We wanted to see if the same type of idea could be applied to model free RL and then found that it worked quite well.
>
> We now give a detailed response to the reviewer's questions below.
>
> **One of the clearest ways to show results that can be trusted is to reproduce existing results from another publication, then outperform those results, but it didn't seem like this was done.**
>
> For experimental results, we address many of the points below.  We demonstrate that we have reproduced the baselines utilized and have found nearly identical results to baseline publications. There may have been some misunderstanding about the specific types of environments we used that lead to the reviewer's initial conclusions. Please let us know if you have further questions.
>
> **The offline RL experimental results also are not clear to me. For example, on halfcheetah your baselines achieve around 4.5k score, but in the BRAC paper, the algorithms achieve around 6k score. Where does the discrepancy come from?**
>
> We invite the reviewer to view the performance metrics in the manuscript that we obtained our offline datasets from, D4RL (Fu, 2020: https://arxiv.org/abs/2004.07219).  We used D4RL for evaluation, which is different from that used in the original BRAC paper.  It is easy to see our baseline results (5.4k) align nearly identically with the reported BRAC results on “halfcheetah-medium”. Our BRAC baseline on “halfcheetah-medium-expert ” is 5.5k, which is actually slightly higher than the baseline reported in D4RL.

---

> > ### Author Response · Authors · 2021-11-19
> > **Reviewer GXWR Response (2)**
> >
> > **Also, in the antmaze results in Table 10, your baselines achieve ~0, whereas the in the original paper by Florensa et al. the baseline algorithms reached around 0.6 reward.**
> >
> > The paper referenced by the reviewer relies on a hierarchical architecture for the baselines, along with ways to express the task as a goal-reaching task. These domain-specific modifications can improve performance on AntMaze. Our baselines are domain-agnostic, and it is clear from other publications (Nachum, 2018: https://arxiv.org/abs/1805.08296, Nachum, 2019: https://arxiv.org/pdf/1810.01257.pdf, Li, 2019: https://arxiv.org/pdf/1910.04450.pdf) that the domain-agnostic baseline score is in fact 0.
> >
> > **The experiments with partial observability were based on the work of Yang & Nachum, (2021), but it appears executed in a different way, and it's not easy to see that the baselines reproduced any results from their work.**
> >
> > We will make a couple of clarifications here.  For the environments tested, we randomly zero-masked a dimension of the environment, corresponding with the procedure in the Yang and Nachum work (we verified with the publication’s codebase).  For the baselines in the Yang and Nachum paper, the y-axis in the figures is based on the “proportion” of the optimal agent result (which in Hopper for example is roughly 10% of 4000, which corresponds to our reward value of 400).  Recall that our learning curves only show the first 500k steps.  For Hopper and Walker2d, the results from this work roughly correspond with our results.  For Ant, the discrepancy is due to the fact that we zero out an extra dimension since Ant has a much larger state dimension than the other MuJoCo environments. We did this to make the problem slightly more difficult so we could better showcase our improvements (it is clear from the Yang/Nachum results that zeroing out a single dimension only negligibly affects the difficulty).  For HalfCheetah, our reward is slightly higher than the reported baseline. HalfCheetah can have high variance results depending on the network initializations.  We also found that our results were high variance (the standard error across all seeds on all timesteps was ~1532 for baseline SAC) and believe this was the cause of the discrepancy.
> >
> > **The Bsuite umbrella-length experiment seems fine. The result of ~0.6 is higher than the result of ~0.5 for DQN reported in the bsuite paper. However, the new baselines in this paper (PPO and SAC) only had a score of ~0.4. Also, why were none of the other bsuite environments tested?**
> >
> > We have updated the paper to include the standard DQN baseline in addition to a DQN with PGIF architecture for the bsuite experiment, exhibiting an improvement with PGIF.  As for other bsuite environments being tested, there are very few credit assignment centric ones.  The other rain/shine credit assignment task is umbrella-features, where the number of features in each state is varied. Since our method deals with incorporating long term future information, we decided that varying the length of the trajectory would be a more interesting and fitting experiment. As for the discounting chain task (another credit assignment task dealing with a discounting horizon), we will plan to evaluate it and include results in the final draft of our work.
> >
> > **In minigrid there is also no reference to any previous publication showing the same results as produced in this paper.**
> >
> > For MiniGrid, there are limited publications with experimental results on the credit assignment environments we utilized.  We will make our code fully available in hopes of producing a reproducible benchmark using PPO on these environments.  We did find a paper using a DQN (Co-Reyes, 2021: https://arxiv.org/pdf/2101.03958.pdf) and found that for environments like DoorKey and Unlock, the reward is very low with the standard DQN baseline.  As for Door-Key, we found an A2C baseline that roughly corresponds to our PPO performance (https://arxiv.org/pdf/2103.14823), keeping in mind the referenced work does not appear to be peer reviewed.

---

> > > ### Author Response · Authors · 2021-11-19
> > > **Reviewer GXWR Response (3)**
> > >
> > > **It is quite possible that the experiments are performed solidly; however, this is not explained sufficiently for me to understand this. I would appreciate if the authors can comment on the correctness of their implementations, and provide evidence that the results achieved by their baseline algorithms are competitive (e.g., by providing references to papers where the scores are similar to the baselines implemented in this paper).**
> > >
> > > Again, we invite the reviewer to examine the baselines that we have reproduced in offline RL (D4RL, Fu, 2019: https://arxiv.org/abs/2004.07219).  We also invite the reviewer to examine the original soft actor critic paper (Haarnoja, 2018: https://arxiv.org/abs/1801.01290) where we achieve extremely similar reward curves for the first 500k steps (in Appendix Section E in our work and Figure 1 in the SAC paper).  For MiniGrid, we admit that there are limited publications with experimental results on the credit assignment environments we picked.  We will make our code fully available in hopes of providing a reproducible benchmark using PPO on these environments.
> > >
> > > **I can't think of any good theoretical reason why one would want to do what the authors did in their paper, so I think that the experimental results are crucial to convey usefulness and validity of their approach.**
> > >
> > > Our work was inspired by using Z-forcing for learning models that incorporate the long term future (Ke, 2019: https://arxiv.org/abs/1903.01599).  We wanted to see if the same type of idea could be applied to model free RL and then found it worked quite well.
> > >
> > > As for the conceptual motivation, we aim to allow the agent to incorporate a learned representation of the future for value function estimation.  This representation will contain information about how bad or good the outcome was at an arbitrary temporal distance in the trajectory.  Providing this information at each update to value function estimates can implicitly help the value function understand cause/effect. This can provide an important tool to determine the significance of current actions and states on long-term outcomes.  Imagine a sparse reward setting such as AntMaze,  where the agent only receives a reward of $+1$ if it successfully learns to walk and reaches the goal state.  By incorporating various elements of the future into value functions we can learn a representation of other important aspects of the trajectory such as optimal gait control that do not immediately result in a reward.  It is easy to see how values of states and actions can be better estimated in hindsight in a variety of comparable tasks.
> > >
> > > **The algorithm was motivated from the point of view of allowing to "look into the future" to improve the optimization performance; however, typical policy gradient algorithms already do this when using Monte Carlo returns.**
> > >
> > > We disagree that policy gradient algorithms receive the Monte Carlo returns as input to the policy or value function.  They are purely used to compute losses, such as Bellman or TD errors.  In PGIF, future trajectory information is incorporated using a latent variable directly as input to the policy or value function.
> > >
> > > **The discussion around equations 1 and 2 is not wrong, but I think it would be better to explain that this is a surrogate loss that allows easily implementing the policy gradient using automatic differentiation, and that only the log⁡π term will be differentiated. Originally, the policy gradient theorem was not presented in the way that you did.**
> > >
> > > We have added a small change to our manuscript to incorporate this suggestion.  Also, note that in the line before this equation, we clarify that there are no passing gradients through the $\rho_\pi$ term.

---

> > > > ### Author Response · Authors · 2021-11-19
> > > > **Reviewer GXWR Response (4)**
> > > >
> > > > **On page 5 in the VPN section, what do you mean by predicting the discounts? Isn't the discount factor just a constant? Also, is the encoding network trained end-to-end?**
> > > >
> > > > The discount factor in VPNs includes termination of the environment and must therefore be predicted, along with the immediate reward.  The encoding network is not trained end to end.
> > > >
> > > > **The figure axis labels are small and hard to read. Also, you used a non-vector figure format, which makes them blurry. I prefer vector format figures so that they look sharp when you zoom in. Moreover, it would be better to use more than color to differentiate between the lines.**
> > > >
> > > > This is a good suggestion and we have made changes for the partially observable MuJoCo environment results.  We will incorporate this suggestion for all figures including the ones in the appendix for the final version.  We will use vector figures in our final version.
> > > >
> > > > **"Furthermore, we find that our method is much slower to train than the baselines we compare to, due to the fact that the architecture requires training an LSTM, with at times, very long trajectories as input." It is good that you mentioned this, but can you roughly quantify it? How much slower was it?**
> > > >
> > > > We would like to emphasize that using a transformer as opposed to an RNN greatly reduces the speed gap with minimal effect on final return (as shown in our experiments in Appendix G). Still, when using LSTM as described in the main text, speeds were roughly 2x slower for MiniGrid and 3-5x slower for MuJoCo.
> > > >
> > > > In addition, we thank the reviewer for noticing other small errors in the text and have fixed these.

---

> > ### Comment · Reviewer_GXWR · 2021-11-25
> > **Update after author rebuttal**
> >
> > I have updated my review and increased the score to 6.
> > I also noted some remaining concerns.
> > Please comment here if you have any thoughts on my remaining concerns.

---

### Official Review · Reviewer_UbCM · 2021-10-31

**Correctness:** 3
**Technical Novelty And Significance:** 2
**Empirical Novelty And Significance:** 2
**Recommendation:** 6
**Confidence:** 5

**Main Review:**

Strengths:
1) The paper attempts to address the problem of credit assignment in RL which is very important. The authors build off of the existing idea of learning future conditioned policy/Q function and introduce several tricks (informational bottleneck and z-forcing) to make it work.
2) The paper demonstrates that the proposed approach can be applied to several RL algorithms, such as PPO, SAC, and BRAC.
3) The paper features an extensive empirical study across several tasks that feature various challenge, such as sparse reward credit assignment, partial observability, and learning from offline datasets.

Weaknesses:
1) The novelty of the method is limited as it largely builds off of existing work from Harutyunyan et al, 2019 and Mesnard et al., 2021. The the authors attempt to differentiate their work from the prior methods, but the difference mostly comes from the fact that PGIF can be made off-policy, unlike the prior work (i.e. Hindsight Credit Assignment) which is on-policy.
2) It is not clear how much information about the future one can force into the prior distribution that is only conditioned on the current state.  I'm not convinced that this does any better than for example SAC (or DDPG/TD3) + n-step returns. In general, the paper omits comparison to any methods that uses n-step return.
3) The authors feature z-forcing as an advantage of their method, but I believe that the need for z-forcing stems from the fact that there is very little mutual information between the prior and posterior distributions, so the backward model without z-forcing will simply collapse to  a trivial solution in order to minimize KL-div to the prior distribution.
4) The final method ends up being extremely complicated featuring several auxiliary losses and the recurrent model. The empirical results are encouraging but not ground breaking. Thus I'm not sure the algorithm is practical and can enjoy wide adoption considering its complexity.

Questions:
1) Re the umbrella-length experiment: what episode length has been actually used? or this presents an average results across multiple horizons?
2) Re the umbrella-length experiment: it would be interesting to see a baseline that uses n-step returns, for example DDPG/TD3 + nstep. It would be much stronger baseline than SAC in these sparse reward/long credit assignment settings.
3) Re the MuJoCo partial-observability experiments: I don't think it is a fair comparison to compare SAC-based PGIF against vanilla SAC (that doesn't use recurrent policy/Q). Obviously PGIF, that has a way to deal with partial observability, will outperform SAC. Instead, the authors compare to PPO-LSTM, which can deal with partial observability, but not as strong baseline as SAC on MuJoCo. It would be interesting to also see performance of SAC-LSTM and PPO-based PGIF to make this experiment valid.

**Summary Of The Paper:**

The paper proposes to incorporate future information for more accurate policy and Q-function estimation, and consequently to derive variants of policy gradient based algorithms such as PPO, SAC, and BRAC. The authors argue that the ability to condition on the future information enables better credit assignment. The authors empirically demonstrate that their approach improves performance on a range of tasks that feature various challenges, including sparse rewards credit assignment, partial observability, and offline RL.

**Summary Of The Review:**

While the paper studies an interesting direction of learning future conditioned policy/Q-function, which is quite exciting, I have several issues with the paper, namely:
1) The incremental nature of the work.
2) The complexity of the method.
3) Several concerns regarding empirical evaluation.

Taking these into account, I'm not sure the paper contributes enough in its current state to justify an acceptance.

---

> ### Author Response · Authors · 2021-11-19
> **Reviewer UbCM Response (1)**
>
> Thanks for closely reading our submission! We address your feedback below. Please let us know if this addresses all your concerns, and if so, whether you will reconsider your overall score.
>
> We will first highlight some of our main rebuttal points, and the additional ablations we have added to our rebuttal revision.
>
> * Concerning the novelty of this work, we are the first practical and scalable deep RL algorithm that incorporates hindsight.  Other works do not experiment in complex high dimensional environments, where nuanced approaches like Z-forcing are necessary to learn useful latent variables.
> * We believe our empirical support for our method is extensive and supports the novelty of it. We evaluate our method on numerous high dimensional environments, dealing with a multitude of problems like offline RL, sparse rewards, partial observability and difficult credit assignment.
> * As for the complexity of our method, it involves two additional auxiliary losses, so we disagree that it is highly complex. All in all, our method is about 192 lines of code on top of the standard PPO model, including extraneous comments and defining/initializing all the relevant networks.
> * As for the practicality and computational time of our method, for the recurrent model, we can simply switch it with a transformer and achieve similar performance with greatly increased speed (Appendix G),
> * For empirical evaluation concerns, we added results showing performance using SAC with n-step rewards on MuJoCo (Appendix I.2), and a DQN with n-step rewards on bsuite (Table 1).  We do not find significant improvements by using n-step rewards, and a degradation of learning in some cases using SAC.
>
> **The novelty of the method is limited as it largely builds off of existing work from Harutyunyan et al, 2019 and Mesnard et al., 2021. The the authors attempt to differentiate their work from the prior methods, but the difference mostly comes from the fact that PGIF can be made off-policy, unlike the prior work (i.e. Hindsight Credit Assignment) which is on-policy.**
>
> While the reviewer presents a good point on the motivation being similar to other works, we would like to note that none of the existing works are practically useful in highly complex environments that require finely tuned deep RL methods. Our experimentation deals with numerous complex and high-dimensional environments, such as MuJoCo and MiniGrid in addition to conceptually simpler Bsuite environments.  Mesnard et al. evaluates its method on low dimensional toy contextual bandit environments, color mapping tasks and very simple gridworlds.  The environments in Mesnard et al. are also not publicly available. Harutyunyan et al, 2019 does not utilize deep RL in any environment.  Our work is thus the first to implement hindsight in a practical and scalable way.  We show improvements in both standard online and offline RL, as well as in environments that specifically test partial observability and credit assignment, and the algorithmic details introduced by PGIF are crucial to attaining these results. For example, existing “Hindsight Credit Assignment” works do not employ Z-forcing, while ablations in our paper make it clear that Z-forcing is crucial to show improvement and avoid collapse of training.  Furthermore, other hindsight methods have not open sourced their codebase, which we will do.
>
> **It is not clear how much information about the future one can force into the prior distribution that is only conditioned on the current state. I'm not convinced that this does any better than for example SAC (or DDPG/TD3) + n-step returns. In general, the paper omits comparison to any methods that uses n-step return.**
>
> We have included experiments using SAC + n step returns on a few MuJoCo environments to highlight the differences in Appendix I.2.  There is, in fact, a degradation of performance when using higher values of $n$.  We also note that in PPO, we use GAE-$\lambda$ to estimate advantage values, which utilizes lambda returns.
>
> **The authors feature z-forcing as an advantage of their method, but I believe that the need for z-forcing stems from the fact that there is very little mutual information between the prior and posterior distributions, so the backward model without z-forcing will simply collapse to a trivial solution in order to minimize KL-div to the prior distribution.**
>
> This is true,  Z-forcing is a requirement to ensure latent variables incorporate relevant information. We reference papers discussing this phenomenon in our introduction now, such as (Bowman, 2015: https://arxiv.org/abs/1511.06349, Chen, 2017: https://arxiv.org/pdf/1611.02731.pdf).

---

> > ### Author Response · Authors · 2021-11-19
> > **Reviewer UbCM Response (2)**
> >
> > **The final method ends up being extremely complicated featuring several auxiliary losses and the recurrent model. The empirical results are encouraging but not ground breaking. Thus I'm not sure the algorithm is practical and can enjoy wide adoption considering its complexity.**
> >
> > Our method involves two additional auxiliary losses, so we disagree that it is highly complex.  As for the recurrent model, we can simply switch it with a transformer and achieve similar performance with greatly increased speed, indicating that our method is robust to implementation details.  All in all, our method is about 192 lines of code on top of the standard PPO model, including extraneous comments and defining/initializing all the relevant networks. We also run numerous ablations on our method to show the importance of the auxiliary losses.
> >
> > **Questions:**
> >
> > **Re the umbrella-length experiment: what episode length has been actually used? or this presents an average results across multiple horizons?**
> >
> > The number of the random information “distractor” states (corresponding to horizon) in the episode is varied, with the average result reported. Specifically, there are 23 different umbrella length environments, each with a different number of distractor steps. Each one is treated as a separate experiment with its own training and evaluation.  The average regret over all these different length-varied experiments is then reported and normalized.
> >
> > **Re the umbrella-length experiment: it would be interesting to see a baseline that uses n-step returns, for example DDPG/TD3 + nstep. It would be much stronger baseline than SAC in these sparse reward/long credit assignment settings.**
> >
> > This is a useful suggestion. We added a DQN baseline (which is already implemented with the Bsuite codebase) with n-step returns and achieved a similar reward to the original Bsuite paper.  We note that MiniGrid is using GAE-$\lambda$ to estimate advantage values, which utilizes a lambda return.  As for DDPG and TD3, we are quite limited by the Bsuite codebase in our ability to add multiple additional algorithms, so we selected PPO since it does in fact utilize a lambda return.  It may be interesting for the final version of the paper to look at DDPG with n-step returns as opposed to DQN.
> >
> > **Re the MuJoCo partial-observability experiments: I don't think it is a fair comparison to compare SAC-based PGIF against vanilla SAC (that doesn't use recurrent policy/Q).**
> >
> > We emphasize that SAC+PGIF also does not use a recurrent policy/Q. The recurrent network is only used during training to process *future* information. During inference, SAC+PGIF and SAC both select actions based only on the current state and do not utilize any past information.

---

> > > ### Comment · Reviewer_UbCM · 2021-11-25
> > > **Update to rating**
> > >
> > > Thank you for a very informative and detailed response. In the light of new information and additional experiments I've updated my initial rating.

---

### Official Review · Reviewer_EPSM · 2021-11-01

**Correctness:** 4
**Technical Novelty And Significance:** 3
**Empirical Novelty And Significance:** 3
**Recommendation:** 6
**Confidence:** 3

**Main Review:**

# Method
- I see why the proposed method would help in partially observed environments: conditioning on some representation of the future allows the policy to condition on a function of the hidden information and give a lower variance policy gradient. However, in fully observed online settings it is not obvious to me why this method improves performance, and I don't see much intuition provided in the paper.
- Equations 4 and 5: why are $u$ and $z$ separate? Did you try learning a single latent variable for both policy and $Q$ function?
- This latent variable $u$ seems to contain information about both the dynamics and logging policy since these interact to form the future. If we know the logging policy, the information bottleneck means we artificially throw away information that we know. This seems like an undesired side-effect of this information bottleneck.
- 3.2: "Especially in settings with highly stochastic learning signals". What does this mean?
- I didn't follow how $p_{v^{(u)}}$ was updated. It is described as a "learned prior" but it only appears in the KL term in the loss. If the network for learning this has sufficiently many parameters will it not just overfit to tracking $q_{\phi}$?

# Experiments
- It seems that Table 1 might address partial observability. If the first state and action are not maintained in the state for the rest of the task (if I have misunderstood this please clarify), we do not have an MDP. If this is true, I don't see a convincing case for this method improving performance much outside of partially observable environments (in online RL) besides the minor improvement in appendix E, and the great results in appendix H. If the focus of improvement is in partially observable or offline settings, this should be made clear in the motivation.
- In Appendix H, I did not find the explanation for improvement convincing, even though the results themselves are impressive. "We hypothesize our algorithm has benefits in these environments since as soon as it obtains a reward signal it can adapt quickly and make use of the signal by incorporating it in both policy and value optimization, therefore accelerating learning." With enough Bellman updates, SAC should also be able to propagate this information back to states earlier in the trajectory. How many rounds of Bellman updates are being used in SAC? Is it possible that in fully observed environments, PGIF is giving an optimization benefit by skipping this reward information directly to early-trajectory states instead of indirectly through Bellman updates?
- Figure 2: main text should highlight LavaGapS5 as a case where we don't see improvement. Is there some intuition as to why?
- Section 5.1 is supposed to evaluate the method on sparse rewards, testing credit assignment. But a partially observed environment is used for some of the experiments in this section. Partial observability is the focus of 5.2.
- Table 1 shows great results when using PGIF-SAC, but then in the next experiments in Figure 2 the PPO version is used?
- Why is HIMO not compared to in Table 1? Same for Figure 3. It is explained in the appendix but could perhaps be a footnote in the main paper.
- 5.2: "more realistic scenario" is probably not an accurate way to describe this particular modification to the environment.
- Experiments in appendix E. The environment has no partial observability but we are in a finite horizon setting, meaning that the number of remaining steps in an episode is not known to the agent. It could be that the slight improvement in performance over SAC is because PGIF can provide the policy with the number of remaining steps. This can be tested by including the number of remaining episode steps into the state and rerunning the experiment.
- How does the method perform with no Z-forcing? This seems like a straightforward ablation.
- Mesnard et al. (2021) is a very similar approach that is not compared to.
Why is the improvement in offline continuous control much greater than the online equivalent? Is it because in the offline case the future contains information about the logging policy?

# Minor points
- The proposed method is called PPO-PGIF in the text but in Figure 2 the legend uses PGIF-PPO.
- Appendix B: " Attention augment agents utilize dynamics" Typo
- "PGIF agent is able to efficiently" typo
- Row 1 of Table 7 in the appendix: typo?
- The text on figures (eg legends) is too small.

I'm not too familiar with the related work on hindsight, so my review takes the paper's comparison with this prior work at face value.

**Summary Of The Paper:**

The paper proposes a method for incorporating future trajectory information when training model-free RL methods. Two technical challenges were reducing over-reliance on future information and allowing the policy to make predictions without future trajectory information at deployment time. These are addressed by introducing an information bottleneck. Optimization issues with latent variable models are tackled using two different Z-forcing approaches. The method shows favorable empirical performance in both online and offline settings.

**Summary Of The Review:**

Here I review the key constructive points:
- Beyond the online ant setting in the appendix, I don't see much evidence that this method improves performance outside of partially observable environments in the online setting. The intuition behind why it would help in fully observed environments is also not clear from my reading of the paper. Given this, I think that the paper should focus the motivation more heavily on the POMDP and offline setting. This was the main reason for my score. I'm happy to discuss if I've misunderstood something about the utility of the method in the fully observed setting.
- From my description above I think there are a few ways in which the experiments could be more convincing.

Why I liked the paper and recommended to accept:
The core idea in this paper is very elegant and the variational inference + z forcing to implement this idea is well done. As far as I can tell the use of Z-forcing and the variational bottleneck is novel in this setting. The results on partially observable and offline settings are impressive.

---

> ### Author Response · Authors · 2021-11-19
> **Reviewer EPSM Response (1)**
>
> Thanks for closely reading our submission! We address your feedback below. Please let us know if this addresses all your concerns, and if so, whether you will reconsider your overall score.
>
> We first would like to emphasize a few points regarding the concerns around partial observability and the validity of our experiments.  We deal with the question and other main rebuttal points:
>
> **I don't see much evidence that this method improves performance outside of partially observable environments in the online setting.**
>
> * We find significant improvements in offline RL with full observability.
> * Bsuite has elements of partial observability, but again is considered a standard baseline for testing credit assignment. We highlight our PPO-LSTM baseline in this setting; the use of an LSTM on the Umbrella task makes the task fully observable.
> * We concede that MiniGrid is also partially observable. Nevertheless, the principal difficulties that these experiments are designed to test are credit assignment and sparse rewards.  MiniGrid is a standard baseline for sparse rewards (Raileanu, 2020: https://arxiv.org/pdf/2002.12292.pdf, Zhang, 2021: https://openreview.net/pdf?id=CYUzpnOkFJp).  It can be seen from the author's implementation that most MiniGrid environments are solvable without memory (https://github.com/maximecb/gym-minigrid).  We compare against PPO-LSTM to show a memory baseline, which further supports our conclusion.
> * As for making the experiments more convincing, we have shown an ablation removing Z-forcing (Appendix I.4) and an experiment showing that increasing the number of updates to SAC per training step does not improve performance significantly (Appendix I.3).
>
>
> **I see why the proposed method would help in partially observed environments: conditioning on some representation of the future allows the policy to condition on a function of the hidden information and give a lower variance policy gradient. However, in fully observed online settings it is not obvious to me why this method improves performance, and I don't see much intuition provided in the paper.**
>
> As for the conceptual motivation, we aim to allow the agent to incorporate a learned representation of the future into the value function or policy.  This representation will contain information about how bad or good the outcome was at an arbitrary temporal distance in the trajectory.  Providing this information at each update to value function estimates can implicitly help the value function understand cause/effect. This can provide an important tool to determine the significance of current actions and states on long-term outcomes.  Imagine a sparse reward setting such as AntMaze,  where the agent only receives a reward of $1$ if it successfully learns to walk and reaches the goal state.  By incorporating various elements of the future into value functions we can learn a representation of other important aspects of the trajectory such as optimal gait control that do not immediately result in a reward. We can tune how much access the agent has to this information and increase it over time using the weights of our various KL and auxiliary losses.
>
> **Section 5.1 is supposed to evaluate the method on sparse rewards, testing credit assignment. But a partially observed environment is used for some of the experiments in this section. Partial observability is the focus of 5.2.**
>
> MiniGrid is partially observable, but the main challenge is not overcoming partial observability.  Most MiniGrid tasks can be solved without a memory LSTM as stated in the repository (https://github.com/maximecb/gym-minigrid).  We would argue that MiniGrid is difficult due to the extremely sparse reward, therefore we believe it is more fitting in the section on sparse rewards, but also showcases our method as robust to both challenges in the same environment.

---

> > ### Author Response · Authors · 2021-11-19
> > **Reviewer EPSM Response (2)**
> >
> > **It seems that Table 1 might address partial observability. If the first state and action are not maintained in the state for the rest of the task (if I have misunderstood this please clarify), we do not have an MDP. If this is true, I don't see a convincing case for this method improving performance much outside of partially observable environments (in online RL) besides the minor improvement in appendix E, and the great results in appendix H. If the focus of improvement is in partially observable or offline settings, this should be made clear in the motivation.**
> >
> > The Umbrella task in Bsuite is explicitly designed by the authors to test credit assignment.  In fact, the most common RL tasks have some element of partial observability, such as MiniGrid.  We argue that the difficulty of the Bsuite task is assigning credit to the first action, given the random distractor information presented afterward. This style of “distractor state” experiment is most prevalent in challenging credit assignment tasks. We further highlight that the incorporation of an LSTM in the PPO-LSTM baseline makes the task fully observable.
> > We agree that the main empirical focus of our method is sparse reward and partial observability.  Given the competitiveness of SAC as a baseline on the standard MuJoCo tasks, we do include our improvements as a part of the manuscript's Appendix (though less important than the results in the main paper).
> >
> > **Equations 4 and 5: why are u and z separate? Did you try learning a single latent variable for both policy and Q function?**
> >
> > We did not try learning a single latent variable for both policy and the value function.  These are learned through separate, disconnected networks.  Intuitively, these representations should be different and contain more fine tuned information used specifically for action selection or value estimation.  Also, one of the signals used to train the backwards RNN is either the policy or value loss, so it makes sense for this network to be separate for both policy and value functions.
> >
> > **3.2: "Especially in settings with highly stochastic learning signals". What does this mean?**
> >
> > We decided to change this term to “sparse learning signals” to better fit with the rest of our experiments.  Highly stochastic refers to Bsuite, where the states are random up until the final state.
> >
> > **I didn't follow how pv(u) was updated. It is described as a "learned prior" but it only appears in the KL term in the loss. If the network for learning this has sufficiently many parameters will it not just overfit to tracking qϕ?**
> >
> > It is correct that the learning signal for this prior comes exclusively from the KL term. However, this does not imply that it will overfit to tracking q_\phi with enough parameters, since $q_\phi$ depends on the future trajectory information via $b^U$, whereas the prior only observes the current state. Thus, it may be impossible for the prior to perfectly match $q_\phi$.
> >
> > **In Appendix H, I did not find the explanation for improvement convincing, even though the results themselves are impressive. "We hypothesize our algorithm has benefits in these environments since as soon as it obtains a reward signal it can adapt quickly and make use of the signal by incorporating it in both policy and value optimization, therefore accelerating learning." With enough Bellman updates, SAC should also be able to propagate this information back to states earlier in the trajectory. How many rounds of Bellman updates are being used in SAC? Is it possible that in fully observed environments, PGIF is giving an optimization benefit by skipping this reward information directly to early-trajectory states instead of indirectly through Bellman updates?**
> >
> > We note that the number of bellman updates per data collection step is the same in SAC and PGIF (One per step).  We experimented with using multiple updates per step for the SAC baseline and found that it does not perform significantly better. We show this result in Appendix I.3.
> >
> > **Figure 2: main text should highlight LavaGapS5 as a case where we don't see improvement. Is there some intuition as to why?**
> >
> > We wanted to include an example of where our method did not show improvement. In LavaGap, there is no significant improvement. We hypothesize that this is because the most common trajectories generated during early training are ones where the agent enters the lava.  There is also no negative reward for entering the lava, and the episode is simply terminated.  This creates a situation where our future representation is highly suboptimal, especially during early training.  We have added this explanation to our main text.

---

> > > ### Author Response · Authors · 2021-11-19
> > > **Reviewer EPSM Response (3)**
> > >
> > >
> > > **Table 1 shows great results when using PGIF-SAC, but then in the next experiments in Figure 2 the PPO version is used?**
> > >
> > > PPO is designed to work with MiniGrid and a publicly available codebase is optimized for it. Also, SAC does not handle discrete actions by default. The known working PPO implementation for use with MiniGrid is also suggested by the MiniGrid author (https://github.com/maximecb/gym-minigrid).  Also, we see that PPO is the standard baseline when evaluating MiniGrid in other publications (Flet-Berliac, 2021: https://openreview.net/pdf?id=_mQp5cr_iNy, Chevalier-Boisvert, 2019: https://arxiv.org/pdf/1810.08272.pdf, Igl, 2019: https://arxiv.org/pdf/1910.12911.pdf, Goyal, 2019: https://openreview.net/pdf?id=rJg8yhAqKm).
> > >
> > > **Why is HIMO not compared to in Table 1? Same for Figure 3. It is explained in the appendix but could perhaps be a footnote in the main paper.**
> > >
> > > Since there is no known public implementation of HIMO on deep RL tasks, we chose one set of experiments (and codebase) to implement it with.  The only HIMO implementation is a rudimentary notebook using synthetic data. Nevertheless, we believed it was an important baseline to compare against, so we attempted a good-faith implementation on one of our experiments.  In general, we do not expect HIMO to perform as well as PGIF,  as the backwards net is not trained end to end, like in PGIF, and without Z-forcing it is very difficult to train this architecture (shown in Appendix I.4).
> > >
> > > **Experiments in appendix E. The environment has no partial observability but we are in a finite horizon setting, meaning that the number of remaining steps in an episode is not known to the agent. It could be that the slight improvement in performance over SAC is because PGIF can provide the policy with the number of remaining steps. This can be tested by including the number of remaining episode steps into the state and rerunning the experiment.**
> > >
> > > Even though the Gym MuJoCo environments are truncated, they are not finite horizon tasks. There is a termination function for each environment based on metrics from the agent gait. Even if an agent is truncated without termination at some step, the discount at that step is left at 1, maintaining the Markovian nature of the task.
> > >
> > > **How does the method perform with no Z-forcing? This seems like a straightforward ablation.**
> > >
> > > This was one of the first ablations done with our method and is included in Appendix I.4.  It is well known that Z-forcing is needed to learn useful latent variables in the presence of an autoregressive decoder. Our method performs poorly without the auxiliary loss, with no improvement at all. There is also a collapse of training in some cases. We show an example of this using one environment in Appendix I.4.
> > >
> > > **Mesnard et al. (2021) is a very similar approach that is not compared to.**
> > >
> > > We are currently attempting a good-faith implementation of this method since there is no publicly available code for it. The environments used in Mesnard et al. are also not publicly available.  So far, we have not found that the method of Mesnard et al. performs well in our evaluations.  There is actually a degradation in learning in many.  We aim to incorporate these results in the final version, as we continue to tune this baseline method to the best of our ability for our environments.
> > > Further, we emphasize that the Mesnard et al. work is only tested in very simple environments, not the complex practical ones we utilize.  It does not leverage Z-forcing, which is essential to learn useful latent variables in complex architectures like the ones we have utilized as shown in Appendix I.4 of our paper.  Instead of comparing with Mesnard et al, we chose to compare against HIMO in a few select environments so we can establish a hindsight baseline.
> > >
> > > **Why is the improvement in offline continuous control much greater than the online equivalent? Is it because in the offline case the future contains information about the logging policy?**
> > >
> > > In offline RL, we also do not deal with the exploration problem.  PGIF is primarily developed to improve credit assignment, and we know that Actor-PGIF can hamper exploration initially in some cases.  If we had to hypothesize, it could be due to the PGIF network inducing a higher level of constraint to behaviours found in the offline dataset, since it incorporates longer term information.

---

> > > ### Comment · Reviewer_EPSM · 2021-11-22
> > > **Follow-up Question: Settings with no Partial Observability**
> > >
> > > Thank you for the detailed response. For settings without partial observability, I am still not understanding what the proposed method gains you that cannot be gained by Bellman updates to the value function. Thank you for the added experiments of SAC with more bellman updates per round (I.3). The weaker performance as number of updates is increased by so little is a bit surprising and I would guess could be a result of not having a large enough replay buffer.
> > > The claim is that the proposed method allows the value function to account for future information in that specific trajectory. With a Markov setup and no partial observability, it should not matter whether we give the policy access to future information in the same trajectory or information about expected future rewards. In fact, I would expect bellman backups to perform better in this setting since they pool across trajectories. How do you explain the results in fully observed settings given this?
> > >
> > > Thanks.

---

> > > > ### Author Response · Authors · 2021-11-22
> > > > **RE: Follow-up Question: Settings with no Partial Observability**
> > > >
> > > > **The weaker performance as number of updates is increased by so little is a bit surprising and I would guess could be a result of not having a large enough replay buffer.**
> > > >
> > > > We use a replay buffer of size 1M, which is enough to fit all collected experience. In general for work on MuJoCo, it is not standard to use multiple updates per training step. As SAC has been aggressively optimized in numerous implementations to work well with MuJoCo, it is therefore likely that others have observed the same conclusion as we have.
> > > >
> > > > **​​With a Markov setup and no partial observability, it should not matter whether we give the policy access to future information in the same trajectory or information about expected future rewards.**
> > > >
> > > > A simplistic thought experiment may help here: Consider training a supervised learning model on imagenet.  We present two scenarios.  In the first, the input to the model is the image and the output is the label.  In the second, the input is a tuple of (image, label) and the output is the label.  In both of these scenarios, we have full observability, but it is obvious that it is easier to learn using the second scenario.
> > > >
> > > > Of course, the PGIF setup is more complex than this. Nevertheless, PGIF is akin to the 2nd scenario in an RL paradigm; we provide additional input with information corresponding to the “label”, i.e., the future effects (return and future states) induced by the chosen action. The policy and value networks can use this privileged information as a “crutch” while they build an understanding of all causal relationships in the environment necessary to solve the task. In contrast, Bellman updates only provide information about the return “output” to train the value function.
> > > >
> > > > We also would like to reference the work of Ke, et al. 2019 (https://arxiv.org/pdf/1903.01599.pdf), which uses Z-forcing for model-based RL in a similar manner to PGIF.  It is obvious that incorporating the future is useful in fully observable MuJoCo imitation learning experiments.  Our work was mainly inspired by this method and we aimed to apply similar ideas to model-free RL.  It was therefore not unusual to expect that incorporating future information into the value function provides a greater level of useful information.  We can also think of our method as an indirect way to learn future dynamics information by learning a representation of the future.  This information on transition dynamics is also expected to provide some benefit when utilized as input to the value function.

---

> > > > > ### Comment · Reviewer_EPSM · 2021-11-25
> > > > > **Analogy does not seem to match up with the method proposed**
> > > > >
> > > > > Re: the supervised learning analogy. I agree with your analogy, but this is not what your method is doing. It is instead the equivalent of training a supervised agent on $(X, Z) \rightarrow Y$, where $Z$ is some information bottlenecked representation of $Y$. I agree this would make predicting $Y$ easier, but at test time you do not use $Z$, and in this case, it is not at all obvious to me that training using $X, Z$ helps.
> > > > >
> > > > > Moreover, I do not find this "crutch" argument convincing- it is very vague. It would be much more convincing if there was a toy example where the Bellman update fails, looking into the future works (which you have) and you can explain actually what information about the future your method is making use of that the Bellman update is not using (this I don't see). The Bellman update only provides information about the future value obtained, I agree, but to update the current policy in a fully observed Markov environment, this should be all you need to know. GXWR seems to be having a similar difficulty to myself in understanding really why this looking into the future should be expected to help.

---

### Official Review · Reviewer_9hJX · 2021-11-01

**Correctness:** 4
**Technical Novelty And Significance:** 3
**Empirical Novelty And Significance:** 3
**Recommendation:** 8
**Confidence:** 3

**Main Review:**

Overall, this is a solid paper, that combines an interesting and novel theoretical idea and convincing theoretical results. The clarity of the technical content is a particular strength, and the authors' clear prose helps to communicate non-trivial theoretical concepts and provides intuition for their design decisions. There are still a few places in which I believe the clarity of the paper could be enhanced that would serve to make the paper both easier to follow and the results more impactful.

The main change I would suggest is to rely less heavily on Appendices to communicate conclusions about the various other activities. For example, while the reference to Appendix.F in Sec. 3.2 is quite informative (and mentions that applying PGIF to the policy and value functions degrades performance), for other content, the reader *must* go to the appendices to  understand the conclusions of supplementary experiments: e.g., appendix G in which Transformers are used instead of an RNN; there should be couple sentences in the main text summarizing those results. While I do not think it is an issue that there is so much content in the appendices---and, in fact, it may be a benefit so that more time is devoted to the clear descriptions of the technical content of the paper---mentions of the appendices containing results (including both Appendix G and E) should be expanded on.

Some other smaller questions and suggestions are as follows:
- [Sec 3.2] It would be worth mentioning that the Force and VPN objectives are separate from one another. It was only by the time that section was over that it was clear that only one would be used at a time.
- [Sec 5.1] Some readers may not be as familiar with the Gym-MiniGrid environments. Mention that the agent is given only given local ego-centric view of its environment, which is what gives rise to the partial-observability.
- [Sec 5.2] While not essential, it would be helpful to understand why Fig. 2 includes results only from the State-based forcing of PGIF. Would the results from VPN look quite different? Its omission is especially odd considering that the VPN results are clearly better for the experiments from Table 2.
- [Sec 6] I appreciate the potential risks highlighted in the Discussion section. It might be helpful to include how much slower "much slower to train" implies. Could the authors include an order-of-magnitude number for the difference in wall-clock training time for one of the experimental environments?


**Summary Of The Paper:**

This paper presents Policy Gradients Incorporating the Future (PGIF), a novel approach to incorporate future information during training to improve performance of model-free RL agents that must overcome the challenges of planning in environments with sparse rewards. The approach involves relying on recent work in "Z-forcing" in which training incorporates information about the future, encoded as a learned latent state that depends on the full not-yet-executed trajectory and state information. The future information helps the agent to learn effectively and overcome challenges associated with credit assignment notorious in partially observed environments; additional losses are added to ensure that the latent sate contains sufficient task-relevant information and that the planner does not rely too heavily on future information. The authors show solid results in a number of challenging environments, outperforming competitive approaches (including PPO) in nearly all of these, showing the effectiveness of the technique in both Online and Offline RL experiments.

**Summary Of The Review:**

This seems to be a solid paper with an interesting and well-executed novel theoretical contribution tackling a difficult problem of relevance to many in the RL and planning-under-uncertainty communities. The experiments are also convincing, and compare against popular RL strategies. I believe that the clarity of the paper could be further enhanced, yet point out that the paper is already well-written and easy to follow.

---

> ### Author Response · Authors · 2021-11-19
> **Reviewer 9hJX Response**
>
> Thanks for closely reading our submission! We address your feedback below.
>
> **The main change I would suggest is to rely less heavily on Appendices to communicate conclusions about the various other activities. For example, while the reference to Appendix.F in Sec. 3.2 is quite informative (and mentions that applying PGIF to the policy and value functions degrades performance), for other content, the reader must go to the appendices to understand the conclusions of supplementary experiments**
>
> We have added references to our appendixes as appropriate in other sections of the paper.  For example, we have added the reference to the appendix section as we mention the transformers experiment.
>
> **It would be worth mentioning that the Force and VPN objectives are separate from one another. It was only by the time that section was over that it was clear that only one would be used at a time.**
>
> Good suggestion, we have added this note while presenting them.
>
> **Some readers may not be as familiar with the Gym-MiniGrid environments. Mention that the agent is given only given local ego-centric view of its environment, which is what gives rise to the partial-observability.**
>
> We agree that this could be more clear and have added it.
>
> **While not essential, it would be helpful to understand why Fig. 2 includes results only from the State-based forcing of PGIF. Would the results from VPN look quite different? Its omission is especially odd considering that the VPN results are clearly better for the experiments from Table 2.**
>
> We apologize for this oversight. We will include VPN results in the final draft.
>
> **I appreciate the potential risks highlighted in the Discussion section. It might be helpful to include how much slower "much slower to train" implies. Could the authors include an order-of-magnitude number for the difference in wall-clock training time for one of the experimental environments?**
>
> We would like to emphasize that using a transformer as opposed to an RNN greatly reduces the speed gap with minimal effect on final return (as shown in our experiments in appendix G). Still, when using LSTM as described in the main text, speeds were roughly 2x slower for MiniGrid and 3-5x slower for MuJoCo.

---

### Decision · Program_Chairs · 2022-01-20

**Decision:**

Accept (Poster)

**Comment:**

This paper presents a new reinforcement learning algorithm for POMDPs that specifically deals with the credit assignment problem. The proposed algorithm consists in using at each time-step t of a training trajectory the subsequent future trajectory that starts at time t+1 as additional inputs to the policy and value networks. Instead of using the trajectories directly, two RNNs are used to encode the trajectories into latent two variables that are then given as inputs to the policy and value networks. A key novel contribution of this work is the use of "Z-forcing" to help the RNNs learn the relevant information. Since future trajectories are not available during testing, a "prior" network is trained to predict the latent variable given a state. During testing, the latent variable is sampled from the network. Empirical experiments on simple simulated environments show that the proposed algorithm outperforms several baselines.

Key issues raised by the reviewers include the complexity of the proposed algorithm, the fact that several interesting results are in the appendix rather than the main paper, and the weakness of certain baselines. The authors responses helped clarify these issues, and additional experiments (such as a comparison to a DQN with n-step value updates) were performed and added to the paper. The reviews are updated accordingly.

In summary, the paper contains several novel ideas in the context of learning in partially observable environments. It is not entirely clear similar effects of the proposed algorithm can be obtained by using simpler tricks, but the evidence provided by the authors supports the claim that the algorithm outperforms several SOTA techniques in the context of POMDPS.